# Unlocking Bevacizumab’s Potential: rCBV_max_ as a Predictive Biomarker for Enhanced Survival in Glioblastoma IDH-Wildtype Patients

**DOI:** 10.3390/cancers16010161

**Published:** 2023-12-28

**Authors:** María del Mar Álvarez-Torres, Carmen Balaña, Elies Fuster-García, Josep Puig, Juan Miguel García-Gómez

**Affiliations:** 1Instituto Universitario de Tecnologías de la Información y Comunicaciones, Universitat Politècnica de Valencia, 46022 Valencia, Spain; elfusgar@upv.es (E.F.-G.); juanmig@ibime.upv.es (J.M.G.-G.); 2Applied Research Group in Oncology (B-ARGO Group), Institut Catala d’Oncologia (ICO), Institut Investigació Germans Trias i Pujol (IGTP), 08916 Badalona, Spain; cbalana@iconcologia.net; 3Radiology Department CDI, Hospital Clinic of Barcelona, 08036 Barcelona, Spain; jpuigmd@gmail.com

**Keywords:** glioblastoma, predictive biomarker, angiogenesis, bevacizumab, rCBV

## Abstract

**Simple Summary:**

This study seeks to assess the efficacy of bevacizumab (BVZ) in the context of glioblastoma treatment, with a particular focus on the identification of a predictive biomarker, rCBVmax. The findings indicate that BVZ demonstrates marked benefits for patients with moderately vascularized tumors, resulting in a substantially extended median survival after tumor progression compared to those who do not receive second-line treatment. The proposed utilization of rCBVmax as a biomarker has the potential to enable personalized treatment decisions, enhancing patient outcomes by guiding the selection of optimal therapy. Additionally, the establishment of a threshold at 7.5 for categorizing patients based on tumor vascularity presents a more refined approach to the selection of second-line treatments. This research holds promise for improving the management of glioblastoma and optimizing treatment strategies for individual patients.

**Abstract:**

Background: Aberrant vascular architecture and angiogenesis are hallmarks of glioblastoma IDH-wildtype, suggesting that these tumors are suitable for antiangiogenic therapy. Bevacizumab was FDA-approved in 2009 following promising results in two clinical trials. However, its use for recurrent glioblastomas remains a subject of debate, as it does not universally improve patient survival. Purposes: In this study, we aimed to analyze the influence of tumor vascularity on the benefit provided by BVZ and propose preoperative rCBVmax at the high angiogenic tumor habitat as a predictive biomarker to select patients who can benefit the most. Methods: Clinical and MRI data from 106 patients with glioblastoma IDH-wildtype have been analyzed. Thirty-nine of them received BVZ, and the remaining sixty-seven did not receive a second-line treatment. The ONCOhabitats method was used to automatically calculate rCBV. Results: We found a median survival from progression of 305 days longer for patients with moderate vascular tumors who received BVZ than those who did not receive any second-line treatment. This contrasts with patients with high-vascular tumors who only presented a median survival of 173 days longer when receiving BVZ. Furthermore, better responses to BVZ were found for the moderate-vascular group with a higher proportion of patients alive at 6, 12, 18, and 24 months after progression. Conclusions: We propose rCBVmax as a potential biomarker to select patients who can benefit more from BVZ after tumor progression. In addition, we propose a threshold of 7.5 to stratify patients into moderate- and high-vascular groups to select the optimal second-line treatment.

## 1. Introduction

The robust angiogenesis and abnormal vasculature in glioblastoma IDH-wildtype (GBM) are defining features of these highly aggressive gliomas [1,2,3,4,5,6]. GBMs represent the most lethal central nervous system (CNS) tumors in adults [7,8]. However, characterizing their vascularity presents significant challenges due to their heterogeneity at both intratumoral [4,5,6,9,10,11,12] and interpatient [10,13,14] levels. Moreover, this vascular heterogeneity in glioblastomas is further accentuated by tumor dynamics, resulting in significant variations in vascularity between newly diagnosed and recurrent glioblastomas [3].

To determine tumor vascularization profiles in a non-invasive manner at an early stage, magnetic resonance imaging (MRI) and dynamic susceptibility contrast (DSC) play a vital role [15]. MRI-DSC provides valuable insights into the microvascular characteristics of brain tumors and encompasses crucial parameters, like blood volume and flow, proving invaluable in diagnosis and treatment planning [16,17]. It also aids in determining tumor grade, distinguishing between tumor recurrence and radiation necrosis, and assessing treatment responses [18,19]. Multiple studies consistently show a strong connection between measurements of relative cerebral blood volume (rCBV) and microvascular structures in various types of glioma tumors [9,11,13,14,15].

The significance of vascularity in tumors is rooted in their imperative need for a continuous supply of nutrients and oxygen to fuel their relentless growth [4,5,6,10]. Tumors endowed with an extensive blood supply tend to exhibit accelerated and more aggressive growth patterns [20,21], which, in turn, are associated with poorer prognoses—a correlation well-documented in the literature [22,23,24,25,26,27]. It is no wonder that tumor vascularity has garnered substantial attention in drug development over recent decades [28,29,30,31,32,33,34,35,36,37,38,39,40,41,42,43]. This pursuit was ignited by the underwhelming outcomes observed with cytotoxic agents. Additionally, GBMs’ rapid vascularization makes them promising for antiangiogenic therapy research.

The go-to antiangiogenic agent for glioblastomas is bevacizumab (BVZ), which received approval in the US BVZ, approved by the FDA in 2009 for recurrent tumor treatment, is a monoclonal antibody that inhibits VEGFR-mediated signaling by binding to VEGF-A. Early studies on recurrent GBMs showed response rates of 28–40% and 6-month progression-free survival rates of 40-50%, a notable improvement compared to prior studies with a median PFS6 of only 15% in recurrent GBMs [28,29,30,31,32]. Encouraged by initial success, multiple trials since 2009 show BVZ benefits in progression-free survival (PFS) but not in overall survival (OS) [28,29,30,31,32]. This disparity raises questions about its utility in primary GBMs. However, the use of this antiangiogenic therapy may prove advantageous as a second-line treatment for specific patient groups characterized by particular tumor vascularity profiles.

Given the vast spectrum of neovascularization processes in GBMs [43,44,45,46], a comprehensive analysis of various vascular patterns becomes imperative when deciding on the most appropriate treatment strategy. Moreover, the ongoing debate surrounding the utility of BVZ in Europe [28,29,30,31,32] underscores the need for a more personalized approach, as it is becoming increasingly clear that not all patients are ideal candidates for antiangiogenic treatment.

Our study is dedicated to identifying a subset of glioblastoma IDH-wildtype patients who could derive substantial benefits from BVZ treatment through the establishment of a preoperative selection criterion. Our key objectives include (1) proposing preoperative rCBV as a valuable biomarker to stratify patients with glioblastoma IDH-wildtype into distinct vascular groups; (2) contrasting the responses to bevacizumab between patient groups with moderate- and high-vascular glioblastoma IDH-wildtype; and (3) determining the vascular subgroup of patients that could reap the maximum benefits from bevacizumab treatment.

## 2. Material and Methods

### 2.1. Cohort Description

This study enrolled a cohort of 106 patients diagnosed with GBM IDH-wildtype [3]. These patients were selected from the extensive GLIOCAT database [47], which consists of individuals from six prominent healthcare centers in Cataluña, Spain. The participating centers included (1) Instituto Catalán de Oncología (ICO) de Badalona (Barcelona), (2) Hospital del Mar (Barcelona), (3) Hospital Clínic (Barcelona), (4) ICO Hospitalet (Barcelona), (5) ICO Girona (Girona), and (6) Hospital Sant Pau (Barcelona). A material transfer agreement was approved by all the participating centers, accompanied by an acceptance report issued by the Ethical Committee of each institution. The inclusion criteria for this study were meticulously defined, necessitating that patients meet the following conditions:

Histopathological confirmation of GBM IDH-wildtype, with diagnoses falling between June 2007 and May 2015.

Accessibility of complete MRI studies at the presurgical stage, including pre- and post-gadolinium, T1-weighted and T2-weighted Fluid-Attenuated Inversion Recovery (FLAIR), and dynamic susceptibility contrast (DSC) T2*-weighted perfusion sequences.

Inclusion in one of two cohorts: the BVZ cohort (comprising patients treated with bevacizumab after tumor progression) or the control cohort (consisting of patients who did not receive additional treatment after tumor progression);A minimum survival period of 30 days;Compliance with the standard Stupp treatment protocol.

The study employed the RANO criteria to define tumor progression [47]. Patients who were still alive at the time of data analysis were considered as censored observations. The date of censorship was determined as the last contact date with the patient, or if such information was unavailable, the date of the most recent MRI examination.

### 2.2. Magnetic Resonance Imaging (MRI)

Prior to surgery, each patient underwent standard-of-care MR examinations, which encompassed a pre- and post-gadolinium-based contrast agent-enhanced T1-weighted MRI, as well as T2-weighted FLAIR and DSC T2* perfusion MRI scans. A sole set of DSC-MRI images, obtained within a single imaging session, were utilized to compute the relative cerebral blood volume (rCBV) for individual patients (refer to Appendix A).

### 2.3. MRI Processing and rCBV Calculation

To process the MRI data and compute vascular markers, we utilized ONCOhabitats [18] (www.oncohabitats.upv.es, accessed on 1 July 2022), an automated unsupervised method designed to characterize the heterogeneity of enhancing tumor and edema tissues at both morphological and vascular levels while deriving vascular biomarkers. This method consists of four key stages (Figure 1):

1. MRI Pre-processing: This phase includes voxel isotropic resampling of all MR images, correction of magnetic field inhomogeneities and noise, rigid intra-patient MRI registration, and skull stripping;

2. Tissue segmentation of glioblastomas: Achieved through an unsupervised segmentation method that employs a state-of-the-art deep learning 3D convolutional neural network (CNN), taking T1c, T2, and FLAIR MRIs as inputs;

3. DSC perfusion quantification: In this stage, we compute biomarkers like rCBV maps, relative cerebral blood flow (rCBF), or Mean Transit Time (MTT) for each patient. T1-weighted leakage effects are automatically corrected using the Boxerman method, while gamma-variate curve fitting corrects for the T2 extravasation phase. rCBV maps are generated by numerically integrating the area under the gamma-variate curve;

4. Hemodynamic Tissue Signature (HTS) habitats: The HTS provides an automated unsupervised method to characterize the heterogeneity of enhancing tumor and edema tissues in terms of the angiogenic process within these regions. Four sub-compartments for GBM are identified: two within the active tumor, High Angiogenic Tumor habitat (HAT) and Low Angiogenic Tumor habitat (LAT), and two within the edema, Infiltrated Peripheral Edema habitat (IPE) and Vasogenic Peripheral Edema habitat (VPE).

For a more in-depth understanding of the ONCOhabitats method, please refer to [7,19]. The method’s multisite robustness and prognostic capability were validated in an international multicenter study, and the results were published in [8].

### 2.4. rCBV Threshold to Define Vascular Groups

To stratify patients according to tumor vascularity, we used the maximum relative cerebral blood volume (rCBV_max_) calculated at the HAT habitat (Figure 1) since our previous studies shown it to be the most relevant prognostic marker calculated with the ONCOhabitats method, and it was used in previous studies to define the vascular groups [7,8,25,45]. The optimum cutoff threshold was determined by the C-index method [8,13].

### 2.5. Statistical Analyses

#### 2.5.1. Dataset Description: Distinctions between Moderate- and High-Vascularity Groups

We provided a comprehensive overview of key demographic, clinical, and molecular factors for the complete cohort and for the stratified groups, specifically the moderate-vascular and high-vascular groups, as well as those who received BVZ second-line treatment and those in the control group. The variables examined for each subgroup included gender, age at diagnosis, overall survival duration, extent of tumor resection, and MGMT methylation status.

#### 2.5.2. Analysis of Survival Differences among Groups

First, to analyze the global benefit of providing BVZ for the entire cohort and then each vascular group, Kaplan–Meier curves were performed. In this analysis, we assessed differences in survival from tumor progression to exitus depending on whether BVZ was provided, and tumor vascularity. A log-rank test was used to determine any statistical differences between the estimated survival functions of the vascular populations. The number of patients included in each group, the median survival times from the progression of each group, the differential survival times, and the *p*-values are reported.

#### 2.5.3. Comparison of Responses to BVZ between Vascular Groups

To compare the response to BVZ between the two vascular groups, basic metrics related to survival were defined. We compared the proportions of patients from each group with survival from tumor progression at 3, 6, 12, 18, and 24 months.

## 3. Results

### 3.1. Benefit of Providing BVZ for the Entire Cohort

Given that BVZ was administered after tumor progression, we assessed survival differences by analyzing the time from progression to exitus, measured in days. The median survival from progression to exitus for the entire cohort was 114 days. However, after conducting a log-rank test, we observed significant differences between patients treated with BVZ and those who did not receive treatment (265 days vs. 61 days from progression to exitus, respectively). These disparities in survival are visually depicted in Figure 2.

### 3.2. Cohort and Group Description

This study included a total of 106 patients diagnosed with glioblastoma IDH-wildtype and treated with the standard Stupp treatment. The entire cohort was stratified into two groups based on tumor vascularity, as determined by rCBVmax in HAT. An optimal cutoff value of 7.5 was used. Patients with rCBVmax in HAT < 7.5 were categorized into the moderate-vascular group, while those with rCBVmax in HAT > 7.5 were placed in the high-vascular group. Appendix A showcases three MRI examples for each vascular group.

Table 1 provides an overview of the primary demographic, clinical, and molecular variables for the entire cohort and the vascular groups. Statistical analysis revealed non-significant differences between these groups.

Out of the 106 patients, 39 received second-line treatment with the antiangiogenic agent BVZ following tumor progression, while the remaining 67 patients did not receive any additional treatment. Table 2 summarizes the data concerning demographic, clinical, and molecular variables for these two groups, along with the corresponding statistical comparisons.

Figure 3 showcases a selection of rCBV maps and vascular habitats, processed using the ONCOhabitats method, for both the high-vascular and moderate-vascular groups.

When considering both vascularity and second-line treatment, the cohort was further divided into four distinct groups:

Moderate vascularity with second-line BVZ treatment: 11 patients;

High vascularity with second-line BVZ treatment: 28 patients;

Moderate vascularity without second-line BVZ treatment: 14 patients;

High vascularity without second-line BVZ treatment: 53 patients.

### 3.3. Survival Differences between Vascular Groups

To assess variations in survival times across the vascular groups, Mann–Whitney tests were conducted. Figure 4 illustrates boxplots depicting the disparities in survival from progression to exitus among the four groups, which take into account tumor vascularity and second-line treatment. The respective *p*-values derived from the Mann–Whitney tests are presented to highlight statistically significant differences.

Notably, among the group of patients receiving BVZ treatment, a significant discrepancy in survival was observed based on the initial vascularity measurement at the high angiogenic tumor habitat. Patients with lower initial vascularity levels demonstrated significantly longer survival times following BVZ administration.

Furthermore, Kaplan–Meier curves were generated for the four groups (Figure 5), revealing substantial distinctions in the benefits conferred by BVZ depending on whether the patient had a moderate-vascular or high-vascular tumor. Patients with high-vascular tumors experienced a median survival benefit of 173 days when comparing median survival times between those who received BVZ treatment and those who did not (median survival of 235 and 62 days, respectively; log-rank *p*-value: 0.0027). In contrast, patients with tumors displaying moderate vascularity and undergoing BVZ treatment had a survival period that was 306 days longer than those with moderate vascularity who went untreated in the second line (median survival of 357 and 51 days, respectively; log-rank *p*-value: 0.0014). The survival benefit following BVZ second-line treatment was notably more pronounced for patients with moderate tumor vascularity, with a survival difference that was nearly twice as long (173 vs. 306 days) compared to untreated patients.

### 3.4. Enhanced Responses to BVZ in Patients with Moderate-Vascular Tumors

To assess the responses to BVZ treatment in the two vascular groups, we established key survival metrics: survival from tumor progression at 3, 6, 12, 18, and 24 months. The percentages of patients achieving these specific survival durations after BVZ administration are detailed in Table 3.

It is noteworthy that only 14.3% of patients with high-vascular tumors achieved a full 3 months of survival following BVZ treatment, which failed to realize a significant survival benefit. Furthermore, merely 28.6% of patients with highly vascular tumors managed to reach the one-year survival mark, with no patients reaching the two-year milestone. In contrast, more favorable responses were observed in the moderate-vascularity group when BVZ treatment was administered after progression.

## 4. Discussion

The debate surrounding the use of bevacizumab as a second-line treatment for patients with glioblastoma IDH-wildtype has persisted for over a decade [47]. The uncertainty in survival benefits stems from varying results across different clinical trials [29,32,38,39,40,41,44].

In this study, we set out to address three primary objectives: (1) analyze the benefits of administering BVZ following tumor recurrence in a retrospective, multicenter cohort of 39 patients (compared to 67 control patients) with glioblastoma IDH-wildtype; (2) assess the significance of tumor vascularity in the benefits derived from BVZ administration; and (3) propose preoperative rCBV as a valuable biomarker for stratifying patients based on their tumor vascularity, providing pertinent insights from the presurgical stages to guide second-line treatment decisions.

For these purposes, we have analyzed differences in survival times from progression to death in the two treatment arms and groups stratified by preoperative rCBV_max_ in HAT calculated with ONCOhabitats. Recent studies have demonstrated the utility of this biomarker in selecting those patients who can benefit from an extended treatment with temozolomide (more than the standard six cycles) [24,45].

A “one-size-fits-all” approach does not work for glioblastoma, and this includes BVZ treatment. Previous clinical trials found limited overall benefits. We advocate for a personalized approach using non-invasive biomarkers, like rCBVmax in HAT, to select patient groups from the early stages. Our study shows that selecting patients based on a threshold (7.5) for moderate vascularity significantly extends survival. Patients with moderate vascularity receiving BVZ lived 10 months longer than those who were untreated, which is substantial given the tumor’s poor prognosis.

It is important to note that considering the proposed biomarker and threshold, only 25 of the 106 patients analyzed were considered to be moderate vascular. The authors speculate that this may be one of the possible reasons why the clinical trials did not yield strong conclusions about the benefit of BVZ for patients with glioblastomas. Only a reduced subset (~25%) of patients benefited the most. Although we have seen a benefit in providing BVZ for patients with high-vascular tumors, it is not as remarkable for patients with moderate tumor vascularity. Therefore, the major effect could be hidden if the entire cohort is analyzed as opposed to the adequate group of patients.

No useful therapeutic target has been identified to distinguish patients who benefit from treatment. While some authors have suggested the potential role of rCBV as a predictive biomarker for bevacizumab response [45,46,47,48], our study takes a step further by proposing the use of preoperative rCBVmax in HAT, which is calculated using an automated and validated method and a specific threshold (7.5). It is essential to emphasize that our results suggest the potential benefit of second-line BVZ treatment in patients whose newly diagnosed tumors exhibit a specific vascularization profile.

A notable limitation of this study is that despite having a sufficiently large cohort of 106 patients, when stratified into four groups, some groups consist of a smaller number of patients. Furthermore, it is essential to consider that the vasculature of tumors may differ between newly diagnosed and recurrent glioblastoma. Analyzing follow-up MRIs could provide more accurate information for patient stratification. Future prospective studies will address these limitations and incorporate additional factors, such as patient age, to validate the accuracy of these biomarkers for the selection of BVZ second-line treatment. Furthermore, in future studies involving larger cohorts, the possibility of including an intermediate vascular subgroup will be analyzed.

## 5. Conclusions

In conclusion, our study proposes a valuable biomarker for the stratification of patients based on their tumor vascularization profile from the presurgical stage. Moreover, we have demonstrated that patients with an rCBVmax in HAT lower than 7.5 exhibit significantly longer survival from progression when BVZ is administered as a second-line treatment. This research opens the door to future prospective studies aimed at validating these findings and assessing the potential of this biomarker for the selection of second-line BVZ treatment. Positive outcomes in these studies could pave the way for a more personalized approach to second-line treatment for patients with glioblastoma IDH-wildtype, ultimately enhancing prognosis and quality of life.

## Figures and Tables

**Figure 1 cancers-16-00161-f001:**
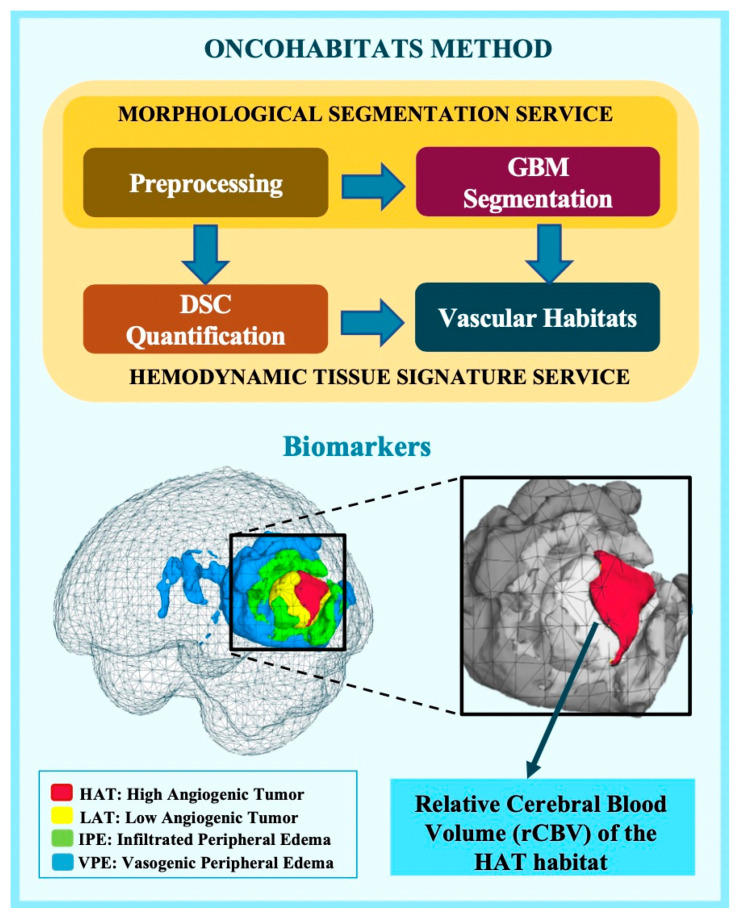
Schematic representation of the ONCOhabitats method, which encompasses four key stages: 1. pre-processing of morphological MRIs (T1, T1c, T2, and flair); 2. segmentation of glioblastoma tissue; 3. quantification of DSC perfusion; and 4. establishment of HTS vascular habitats, including HAT (High Angiogenic Tumor), LAT (Low Angiogenic Tumor), IPE (Infiltrated Peripheral Edema), and VPE (Vasogenic Peripheral Edema). MRI biomarkers, such as the relative cerebral blood volume (rCBV), are obtained from each vascular habitat.

**Figure 2 cancers-16-00161-f002:**
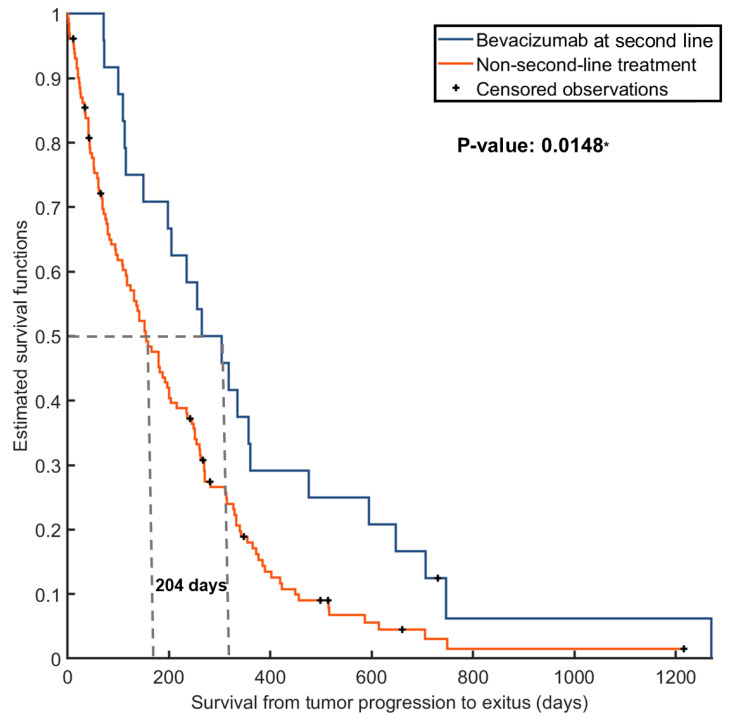
Kaplan–Meier survival curves illustrating significant differences in survival times from progression to exitus based on the provision or absence of bevacizumab treatment after tumor progression. The asterisk indicates significance at *p* < 0.05.

**Figure 3 cancers-16-00161-f003:**
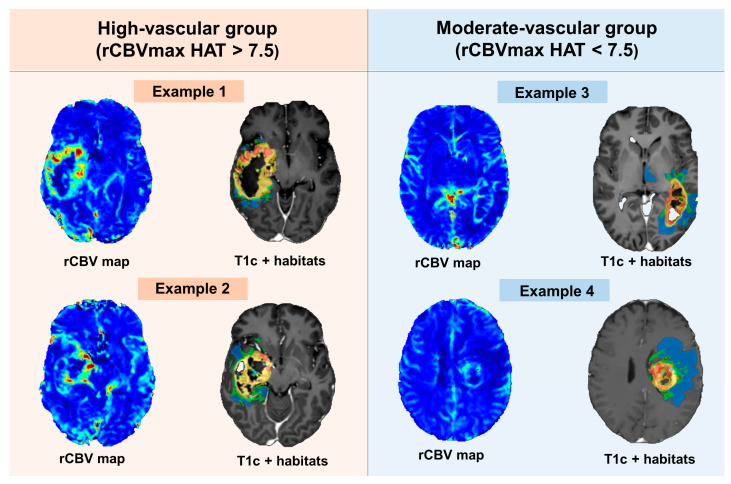
Examples of rCBV maps and vascular habitats delineated by the ONCOhabitats method for the high- and moderate-vascular groups.

**Figure 4 cancers-16-00161-f004:**
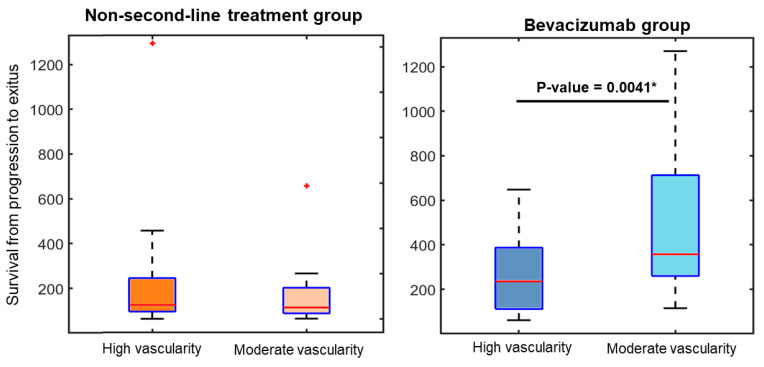
Boxplots revealing substantial variations in survival times from progression to exitus for the group receiving bevacizumab treatment in relation to tumor vascularity. The asterisk in the boxplot represents outlier data points not included within the box.

**Figure 5 cancers-16-00161-f005:**
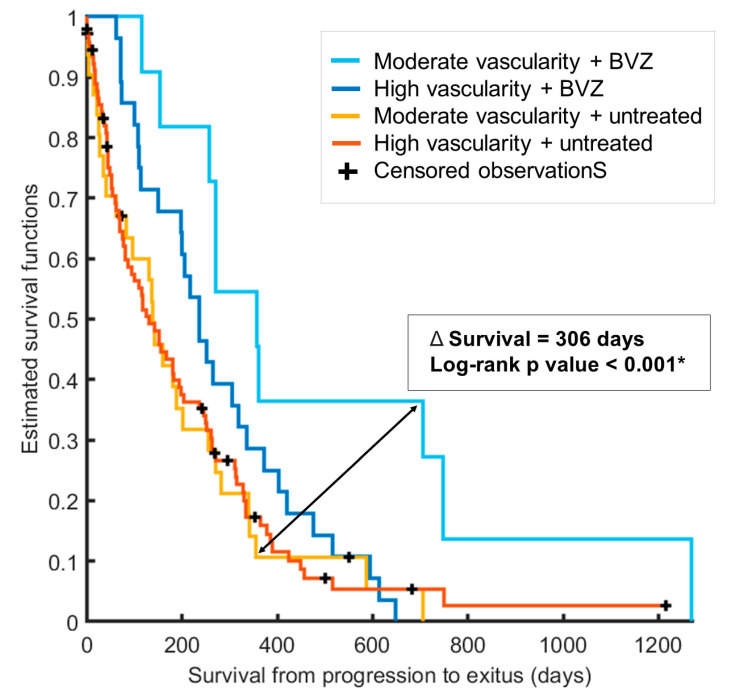
Kaplan–Meier survival curves for the four defined groups based on rCBVmax within the high angiogenic tumor habitat (high or moderate vascularity) and the presence or absence of bevacizumab treatment. These curves highlight significantly extended survival for patients with moderate tumor vascularity who received bevacizumab treatment. The asterisk indicates significance at *p* < 0.05.

**Table 1 cancers-16-00161-t001:** Basic characteristics of the entire cohort and the moderate- and high-vascular groups.

Variables	Entire Cohort	Moderate-Vascular Group	High-Vascular Group	Mann–Whitney Results (*p*-Value)
Number of patients	106	25	81	-
**Proportions per group**	100%	23.5%	76.4%	
**Gender**				
-Number of patients	44	17	27	
-Proportion of females	41.5%	68.0%	33.3%	0.0022 *
**Age at diagnosis (years)**				
-Mean	59	58	59	0.6470
-Range (min, max)	(17,77)	(25,76)	(17,77)	-
**Overall survival (months)**				
-Median	13.4	14.8	13.2	0.5246
**Extent of resection. No. of patients (%)**				
-Complete	20 (18.9%)	6 (24.0%)	14 (17.3%)	0.4586
-Partial maximum	21 (19.8%)	4 (16.0%)	17 (21.0%)	0.5899
-Partial	41 (38.7%)	9 (36.0%)	32 (39.5%)	0.7575
-Biopsy	24 (22.6%)	6 (24.0%)	18 (22.2%)	0.9198
** *MGMT* ** ** methylation status. No. of patients (%)**				
-Methylated	38 (35.8%)	8 (32.0%)	30 (37.0%)	0.6510
-Unmethylated	48 (45.3%)	8 (32.0%)	40 (49.4%)	0.1298
-Unknown info	20 (18.9%)	9 (36.0%)	11 (13.6%)	-
**Preoperative KPS**				
**-Patients with info**	89 (84.0%)	20 (80.0%)	69 (85.2%)	-
-Median KPS	80	80	80	0.8383
**Postoperative KPS**				
-Patients with info	99 (93.4%)	25 (100%)	74 (91.4%)	-
-Median KPS	70	70	70	0.6711
**Treatment. No. of patients (%)**				
**-Complications**	15 (14.1%)	3 (12.0%)	12 (14.8%)	0.7300
-Complete CT	92 (86.8%)	21 (84.0%)	71 (87.6%)	0.6432
**-Complete RT**	99 (93.4%)	23 (92.0%)	76 (93.8%)	0.7555

Complications in surgery; CT: chemotherapy; RT: radiotherapy. The asterisk indicates significance at *p* < 0.05.

**Table 2 cancers-16-00161-t002:** Basic characteristics of groups defined by the second-line treatment (bevacizumab versus control).

Variables	Bevacizumab Group	Control Group	Mann–Whitney Results (*p*-Value)
**Number of patients**	39	67	-
**Proportions per group**	36.8%	63.2%	
**Gender**			
**-Number of females**	15	29	
**-Proportion of females**	38.4%	43.2%	0.6314
**Age at diagnosis (years)**			
**-Mean**	54	61	0.0019 *
**-Range (min, max)**	(17,72)	(25,77)	-
**Overall survival (months)**			
**-Median**	18.3	9.6	<0.0001 *
**Extent of resection. No. of patients (%)**			
**-Complete**	7 (17.9%)	13 (19.4%)	0.8581
**-Partial maximum**	6 (15.4%)	15 (22.4%)	0.3878
**-Partial**	19 (48.7%)	22 (32.8%)	0.1079
**-Biopsy**	8 (20.5%)	16 (23.9%)	0.3028
** *MGMT* ** ** methylation status. No. of patients (%)**			
**-Methylated**	13 (33.3%)	25 (37.3%)	0.6848
**-Unmethylated**	22 (56.4%)	26 (38.8%)	0.0812
**-Unknown info**	4 (10.3%)	16 (23.9%)	0.0862
**Preoperative KPS**			
-Patients with info	37 (94.5%)	52 (77.6%)	-
-Median KPS	80	80	0.2600
**Postoperative KPS**			
-Patients with info	38 (97.4%)	61 (91.0%)	-
-Postoperative	80	80	0.8893
**Treatment. No. of patients (%)**			
**-Complications**	4 (10.3%)	11 (16.4%)	0.3853
-Complete CT	37 (94.9%)	55 (82.1%)	0.0629
-Complete RT	37 (94.9%)	62 (92.5%)	0.6478

Complications in surgery; CT: chemotherapy; RT: radiotherapy. The asterisk indicates significance at *p* < 0.05.

**Table 3 cancers-16-00161-t003:** The proportion of patients after BVZ administration from each vascular group (moderate and high vascularity, respectively) alive at different times from tumor progression (3, 6, 12, 18, and 24 months).

Survival Time from Progression	Moderate-Vascular Group	High-Vascular Group
	Absolute numbers (percentage)	
3 months	11 (100%)	24 (85.7%)
6 months	9 (81.8%)	19 (67.8%)
12 months	4 (36.4%)	8 (28.6%)
18 months	4 (36.4%)	3 (10.7%)
24 months	2 (18.2%)	0 (0.0%)

## Data Availability

Data are available upon request due to privacy restrictions.

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
