# Peer review of "Unlocking Bevacizumab’s Potential: rCBVmax as a Predictive Biomarker for Enhanced Survival in Glioblastoma IDH-Wildtype Patients"

_cancers, 2023, doi:10.3390/cancers16010161_

Round 1

Reviewer 1 Report (Previous Reviewer 2)

Comments and Suggestions for Authors

The authors present their revised manuscript. The addressed the comments adequately. 

Author Response

Thank you very much for your revisions.

Reviewer 2 Report (Previous Reviewer 3)

Comments and Suggestions for Authors

The authors have successfully integrated all suggested improvements in the revised manuscript. The manuscript is now deemed acceptable in its current form.

Author Response

Thank you very much for your revisions.

This manuscript is a resubmission of an earlier submission. The following is a list of the peer review reports and author responses from that submission.

Round 1

Reviewer 1 Report

Comments and Suggestions for Authors

The work submitted by Alvarez-Torres et al proposes a new parameter to select glioblastoma patients for treatment with Bevacizumab, an anti-angiogenic monoclonal antibody. They show that vascularization of tumors, determined as as relative cerebral blood volume (rCBV), differentiates the response of patients. The moderate-vascularity group showed a better response to bevacizumab than the high-vascularity, evidenced by better survival outcomes. In my opinion, this work is adequate for publication.

Author Response

Please, see attached the response letter.

Reviewer 2 Report

Comments and Suggestions for Authors

The authors present the treatment modalities and options with bevacizumab in high grade gliomas. Thereby, the focus on the rCBV as a predictive marker for this treatment. the topic might be of interesting. Although the treatment with bevacizumab is not a standard therapy at all in Europe actually. Nevertheless, the methods sound good and the results are well presented.

However I suggest to address to following aspects:

1. What is new or different in correlation to these manuscripts:

Early post-bevacizumab change in rCBV from DSC-MRI identifies pseudoresponse in recurrent glioblastoma: Results from ACRIN 6677/RTOG 0625.

Boxerman JL, Snyder BS, Barboriak DP, Schmainda KM.Front Oncol. 2023 Jan 26;13:1061502. doi: 10.3389/fonc.2023.1061502. eCollection 2023.   Relative cerebral blood volume as response predictor in the treatment of recurrent glioblastoma with anti-angiogenic therapy. Breda-Yepes M, Rodríguez-Hernández LA, Gómez-Figueroa E, Mondragón-Soto MG, Arellano-Flores G, Hernández-Hernández A, Rodríguez-Rubio HA, Martínez P, Reyes-Moreno I, Álvaro-Heredia JA, Gutiérrez Aceves GA, Villanueva-Castro E, Sangrador-Deitos MV, Alonso-Vanegas M, Guerrero-Juárez V, González-Aguilar A.Clin Neurol Neurosurg. 2023 Oct;233:107904. doi: 10.1016/j.clineuro.2023.107904. Epub 2023 Jul 17. (is this a subgroup?)   2. please add informations about the surgical treatment, radicality, further treatments  Comments on the Quality of English Language

n/a

Author Response

(The authors gave the same response as above.)

Reviewer 3 Report

Comments and Suggestions for Authors

The importance of blood supply in tumors is closely tied to the essential need for a continuous flow of nutrients, crucial for the unrestricted growth of cancer cells. Moreover, ample blood supply often leads to a more aggressive tumor growth pattern, commonly linked with a poor prognosis. The manuscript by Álvarez-Torres et al delves into the exploration of the angiogenic effects of BVZ in patients affected by glioblastoma IDH-wildtype, each exhibiting varying degrees of vascularity. A key innovation of this manuscript lies in the identification of a biomarker, obtained through artificial intelligence, allowing the identification of a subset of glioblastoma patients who could potentially benefit from targeted anti-VEGF therapy. Using ONCOhabitats, the authors automatically calculated relative cerebral blood volume (rCBV) to classify patients according to their vascularity. By distinguishing between moderate and high vascularity groups, the authors administered treatment to both categories. Notably, only patients falling into the moderate vascularity group demonstrated responsiveness to BVZ treatment. The findings presented in the manuscript are of significant interest; however, before it is ready for publication, several refinements are necessary.

Major Comments:

1.      In the introduction, the authors should offer a more thorough explanation of the principles associated with the ONCOhabitats method for calculating rCBV. The linkage between a morphological indicator of angiogenic processes (microvessel density) and the rCBV value should be clarified for better understanding.

2.      It is crucial to specify the precise number of images considered for each patient in obtaining the rCBV value. This clarification is essential for understanding the statistical significance of the method.

3.      The authors should include in the supplementary material the MRI images acquired per patient, which were involved in the calculation of the rCBV value. This should encompass at least three sets for both moderate and high vascularity.

4.      Include in the main text the following reference: Strobel HA, et al. 2021. "Quantifying Vascular Density in Tissue Engineered Constructs Using Machine Learning." Front Physiol. 2021; doi: 10.3389/fphys.2021.650714.

5.      They should engage in speculative discussions on the broader implications of the entire patient cohort benefiting from BVZ treatment. Should the authors consider exploring the possibility of introducing an intermediate class between moderate and high grade?

6.      Investigate whether the MGMT methylation status could represent a genetic condition in the moderate vascularity class that benefits from BVZ treatment. This point is intriguing as it could pave the way for a dual approach involving the rCBV marker, combining Temozolomide and Bevacituzumab.

Minor Comments:

1.      Clarify acronyms upon their initial mention, including in the title (e.g., rCBV, KPS, IDH).

2.      Address the redundant legend for Figure 3 mentioned twice in the text.

3.      Replace Figure 1, which is already published in a prior work by Álvarez-Torres et al in 2021, with another image or a schematic representation.

Author Response

(The authors gave the same response as above.)
